# Potential of *Chlorella* as a Dietary Supplement to Promote Human Health

**DOI:** 10.3390/nu12092524

**Published:** 2020-08-20

**Authors:** Tomohiro Bito, Eri Okumura, Masaki Fujishima, Fumio Watanabe

**Affiliations:** 1Department of Agricultural, Life and Environmental Sciences, Faculty of Agriculture, Tottori University, Tottori 680-8553, Japan; bito@tottori-u.ac.jp; 2Sun Chlorella Corporation, Kyoto 600-8177, Japan; okumura@sunchlorella.co.jp (E.O.); mfujishima@sunchlorella.co.jp (M.F.)

**Keywords:** antioxidants, *Chlorella*, dietary fibers, dietary supplements, folate, lutein, vitamin B_12_, vitamin D_2_

## Abstract

*Chlorella* is a green unicellular alga that is commercially produced and distributed worldwide as a dietary supplement. *Chlorella* products contain numerous nutrients and vitamins, including D and B_12_, that are absent in plant-derived food sources. *Chlorella* contains larger amounts of folate and iron than other plant-derived foods. *Chlorella* supplementation to mammals, including humans, has been reported to exhibit various pharmacological activities, including immunomodulatory, antioxidant, antidiabetic, antihypertensive, and antihyperlipidemic activities. Meta-analysis on the effects of *Chlorella* supplementation on cardiovascular risk factors have suggested that it improves total cholesterol levels, low-density lipoprotein cholesterol levels, systolic blood pressure, diastolic blood pressure, and fasting blood glucose levels but not triglycerides and high-density lipoprotein cholesterol levels. These beneficial effects of *Chlorella* might be due to synergism between multiple nutrient and antioxidant compounds. However, information regarding the bioactive compounds in *Chlorella* is limited.

## 1. Introduction

Microalgae are primarily found in aquatic ecosystems, living in both seawater and freshwater, and are photosynthetic eukaryotic organisms that contain chloroplasts and nuclei, similar to plants. Microalgae more efficiently yield biomass than land-based plants owing to their higher performance in utilizing sunlight and CO_2_, leading to their extremely high growth rates [1]. Therefore, microalgae have been used in the food, pharmaceutical, and cosmetic industries, and their pigments, nutrients, bioactive compounds and whole biomass are already in use worldwide. Recently, various bioactive compounds and nutrients have been detected in both seawater and freshwater microalgae, including cyanobacteria. These compounds and nutrients have been reported to promote human health [1,2]. However, there is limited information regarding the bioactive compounds of freshwater-living *Chlorella* species, which are classified as green algae.

*Chlorella* species can be mass-cultured, and their dietary supplement products are commercially available worldwide. However, the commercial cultivation of their biomass has started only several years ago. *Chlorella vulgaris* was discovered and reported in 1890 by Dr. Martinus Willem Beijerinck, a famous microbiologist and botanist [3]. Another *Chlorella* species, distinguished by the presence pyrenoids in chloroplasts, was identified and accordingly named *C. pyrenoidosa* in 1903 [4]. Since then, more than 20 *Chlorella* species have been characterized, with over 100 strains described [5]. At present, *Chlorella* species are divided into three varieties: *C. vulgaris, C. lobophora, and C. sorokiniana* [6]. *C. sorokiniana* is a sub-species first isolated in 1953 by Sorokin and originally thought to be a thermotolerant mutant of *C. pyrenoidosa* [7,8]. *C. pyrenoidosa*, the subject of many scientific studies, is now called *C. sorokiniana*.

Investigations of the dietary value of *Chlorella* in human health began in the early 1950s, when the use of *Chlorella* as a food source was initiated in the midst of a global food crisis [9]. *Chlorella* was first produced and consumed in Asia, mainly in Japan, and then used as a dietary supplement worldwide [10]. *Chlorella* is produced commercially for use in foods and as a source of its intrinsic compounds. Using large-scale cultivation technology, *C. vulgaris* and *C. pyrenoidosa* are prepared as commercial sources for dietary supplements [11]. Studies have shown that *Chlorella* cells contain a variety of nutrients and bioactive compounds that promote human health and prevent certain diseases [10,12], suggesting that *Chlorella*-derived natural compounds might provide substitutes for synthetic compounds or drugs. The content of natural compounds in *Chlorella* differs greatly between culture conditions and *Chlorella* species [13,14].

Here, we present updated information on the *Chlorella* content of nutrients and bioactive compounds that promote human health. However, at present, there is limited information available regarding the bioactive compounds responsible for its pharmacological activities, which might be due to the synergistic effects of various nutrients and antioxidant compounds in *Chlorella*.

## 2. Nutrients in Commercial *Chlorella* Products

### 2.1. Macronutrients

The macronutrient content of 13 commercially available *Chlorella* products, based on information provided on the packaging label, are summarized in Table 1. Humans cannot digest *Chlorella* cells in their natural state because their cell walls are made of cellulose. Therefore, *Chlorella* cell walls are mechanically broken down in most dietary supplements. An animal study has shown that more than 80% of *Chlorella* proteins are digestible [15].

These *Chlorella* products contain a large amount of proteins (approximately 59% based on dry weight), coinciding with the analytical data of the protein contents of *C. pyrenoidosa* (57%) [16] and *C. vulgaris* (51–58%) [17]. This protein content is higher than that of soybeans (approximately 33%, dry weight). The amino acid composition of *Chlorella* products C and M are shown in Table 2. These amino acid profiles indicate that all essential amino acids for humans (isoleucine, leucine, lysine, methionine, phenylalanine, threonine, tryptophan, valine, and histidine) are present in substantial concentrations in these products. According to the essential amino acid index (EAAI) used to evaluate protein quality for human nutrition, the quality of *C. pyrenoidosa* (EAAI, 1.35) [18] and a commercially available *Chlorella* product (EAAI, 0.92) [19] are higher than that of soybean protein (EAAI, 0.66) [18]. These results indicate that proteins in *Chlorella* products are of high or good quality. Notably, *Chlorella* products contain a considerable amount of arginine (approximately 3200 mg/100 g dry weight), which serves as a substrate for the production of NO, a potent intracellular signaling molecule that influences every mammalian system [20]. Arginine also serves as a potent modulator of immune functions [21].

Approximately 17% (dry weight) of carbohydrates are found in the commercially available *Chlorella* products. Similar results have been reported for *C. vulgaris* [17]. As shown in Table 1, more than 65% of the carbohydrate is dietary fiber, which appears to be derived from the *Chlorella* cell wall. Various polysaccharides have been extracted and characterized [22,23,24,25]. *Chlorella* polysaccharides exhibited a variety of biologically active compounds, including antioxidants [24] and stimulators of plant growth [25]. Tabarsa et al. [26] characterized an immune-enhancing water-soluble α-glucan prepared from *C. vulgaris*.

Commercially available *Chlorella* products contain a small amount of fats (approximately 11%, dry weight) (Table 1), which coincides with the analytical data of the fat content of *C. vulgaris* (14–22%) [17]. *Chlorella* products contain α-linolenic acid (approximately 10–16% of total fatty acids) and linoleic acid (approximately 18% of total fatty acids) but not eicosapentaenoic acid, docosahexenoic acid, or arachidonic acid [19,27]. Approximately 65–70% of the total fatty acids found in commercially available *Chlorella* products are derived from polyunsaturated fatty acids [19,27].

Different growth conditions, such as temperature, nutrient composition, and light availability, can readily alter the levels of biomass, macro- and micronutrients, and other valuable bioactive compounds, including antioxidants, in *Chlorella* cells [28,29,30].

### 2.2. Micronutrients

#### 2.2.1. Vitamins

As shown in Table 3, commercially available *Chlorella* products contain all the vitamins required by humans, i.e., B_1_, B_2_, B_6_, B_12_, niacin, folate, biotin, pantothenic acid, C, D_2_, E, and K, and α- and β-carotenes. *Chlorella* products contain substantial amounts of vitamins D_2_ and B_12_, both of which are well known to be absent in plants. Commercially available *Chlorella* (*C. vulgaris*) products contain higher amounts of folate (approximately 2.5 mg/100 g dry weight) than spinach [31]. Vitamin B_12_ and folate deficiencies induce the accumulation of serum homocysteine, which is involved in cardiovascular diseases. In this section, we discuss vitamin D_2_, vitamin B_12_, and folate.

Vitamin D, a major regulator of calcium absorption, reduces the risk of osteomalacia in adults and rickets in children [32]. The two main dietary forms of vitamin D are vitamin D_2_ and D_3_, which are found in fungi such as mushrooms [33,34] and animal-derived foods such as fish and fish products [35], respectively. Mushrooms have the ability to synthesize ergosterol (known as provitamin D_2_), which is converted into ergocalciferol as vitamin D_2_ upon ultraviolet irradiation [34,36]. Thus, ultraviolet-irradiated mushrooms are suitable for use as vitamin D_2_ sources in strict vegetarians [36]. Cell walls of mushrooms contain high concentrations of ergosterol, which plays a physiological role in modulating cell membrane strength and fluidity similar to cholesterol in animals [37]. Sun-dried, commercially available mushrooms reportedly contain approximately 17 µg of vitamin D_2_ per g dry weight [38]. The bioavailability of vitamin D_2_ from mushrooms has been studied in humans [39,40].

Ergosterol was first reported as the main sterol compound in *C. pyrenoidosa* in the early 1950_S_ [41]. *C. vulgaris* also contains a substantial amount of ergosterol [42,43]. Our unpublished data show that one commercially available *Chlorella* product contains both ergosterol (1.68 mg/g dry weight) and vitamin D_2_ (15.2 µg/g dry weight), similar in amounts to those in sun-dried mushrooms. The vitamin D_2_ in this *Chlorella* product is synthesized from ergosterol upon exposure to sunlight during cultivation (Figure 1). Although it has been reported that vitamin D_3_ is more effective than vitamin D_2_ at increasing the concentration of circulating 25-hydroxyvitamin D [44], *Chlorella* products and sun-dried mushrooms could become sources of vitamin D for vegetarians.

Serum homocysteine (Hcy) is an established biomarker of cardiovascular disease in humans [45,46]. Hcy is a non-protein forming amino acid (Figure 2) produced as an intermediate compound of methionine metabolism and is further metabolized to cystathionine via cystathionine β-synthetase, a vitamin B_6_-dependent enzyme [46]. Alternatively, Hcy can be remethylated back to methionine by methionine synthase, a vitamin B_12_-dependent enzyme. Folate is also required for the remethylation of Hcy by providing 5-methyltetrahydrofolate. Deficiencies in vitamin B_12_ [47], vitamin B_6_ [48], and folate [49] cause hyper-homocysteinemia. Several clinical studies report a correlation between atherosclerosis and deficiencies in vitamin B_12_ and folate [50,51]. Folate deficiency in women before and during pregnancy is associated with neural tube defects in newborns [52]. Plants can synthesize folate compounds de novo, but animals cannot [53]. Thus, plant-derived foods are sources of dietary folates for humans. High concentrations of folate (approximately 1.69–2.45 mg/100 g dry weight) are reported in commercially available *Chlorella* (*C. vulgaris*) products [31], with concentrations similar to those of the products shown in Table 3 (0.3–3.6 mg/100 g dry weight). The main folate compounds identified in *Chlorella* products are 5-CHO-H_4_ folate (60–62%) and 5-CH_3_-H_4_ folate (24–26%) and the minor folate compounds are 10-CHO-folate (5–7%), H_4_ folate (4%), and fully oxidized folate (3–6%) [31]. The chemical structures of the *Chlorella* folate compounds are shown in Figure 3. The main dietary sources of folates are vegetables (25%), bread and cereal products (22%), dairy products (10%), fruit (10%), and oils and fats (5%) [31]. Spinach has high folate content (165 µg/100 g fresh weight; 1.7 mg/100 g dry weight) [31,54], which is similar to that of *Chlorella* products. Thus, *Chlorella* products are an excellent source of folate for humans.

Vitamin B_12_ (B_12_) is synthesized by certain bacteria and archaea but not by plants [55]. Animal-derived foods, such as meats, milk, fish, and shellfish, are the major dietary sources of B_12_ for humans [56]. B_12_ content is high in beef, pork, and chicken livers (approximately 25–53 µg/100 g fresh weight) [56] and in edible bivalves such as clams (approximately 60 µg/100 g fresh weight) [57]. The reported B_12_ content of *Chlorella* products varies from <0.1 to 400 µg per 100 g of dry weight [58,59], consistent with that of the products shown in Table 3 (6–500 µg/100 g dry weight). Among *Chlorella* species, the B_12_ content is much higher in *C. pyrenoidosa* than in *C. vulgaris* when grown under open culture conditions [59]. B_12_ is not essential for the growth of these *Chlorella* species [59,60], suggesting that *Chlorella* cells absorb and accumulate large amounts of exogenous B_12_. Some of the high B_12_-containing *Chlorella* products contain inactive corrinoid compounds such as 5-methoxybenzimidazolylcobamide and cobalt-free corrinoid (Figure 4). Thus, if *Chlorella* products with high B_12_ are consumed as a sole B_12_ source, accurate content estimation requires the identification of B_12_ compounds using liquid chromatography–tandem mass spectrometry [59].

Rauma et al. [61] demonstrated that substantial consumption of *Chlorella* products can supply adequate amounts of B_12_. Another study of strict vegetarians (vegans) with an elevated baseline of serum methylmalonic acid (as an index of B_12_ deficiency) showed that ingestion of 9 g of *C. pyrenoidosa* daily for 60 days resulted in significant decreases in serum methylmalonic acid in 88% of the subjects [62]; serum Hcy decreased and serum B_12_ tended to increase, although the mean corpuscular volume, hemoglobin, and hematocrit levels were unchanged. These results suggest that *Chlorella* products with high B_12_ and without inactive corrinoid compounds are suitable for use as B_12_ sources in humans, particularly vegans.

#### 2.2.2. Minerals

As shown in Table 4, commercially available *Chlorella* products contain a variety of minerals that are required in humans. In particular, *Chlorella* products contain substantial amounts of iron (104 mg/100 g dry weight) and potassium (986 mg/100 g dry weight), of which adequate intake prevents anemia [63] and hypertension [64], respectively. Iron plays physiological roles in respiration, energy production, DNA synthesis, and cell proliferation [65]. The phytates in grains potently inhibit the intestinal absorption of iron because they chelate iron to form an insoluble complex [66]. Thus, people on vegan and vegetarian diets may be at risk for iron-deficiency anemia [63]. Studies in rats and humans have investigated whether *Chlorella* supplementation can prevent iron-deficiency anemia [67,68]. In a cohort of 32 women in the second and third trimester of pregnancy, oral *Chlorella* supplementation (6 g/day) for 12–18 weeks decreased markers of anemia as compared to the control group [68], suggesting that *Chlorella* supplementation significantly reduces the risk of pregnancy-associated anemia.

Selenium (Se) is an essential trace mineral that serves as a fundamental nutrient to human health. It is a component of selenoproteins, such as thioredoxin reductase and glutathione peroxidases, and protects against intercellular oxidative damage [69,70,71]. Therefore, low levels of Se show various pharmaceutical activities, including antitumor and antiaging effects; however, high levels of Se induce the generation of reactive oxygen species. Generally, the organic forms of Se are more bioavailable and less toxic than the inorganic forms of Se. Selenite (SeO_3_^2−^) and selenite (SeO_4_^2−^) are the predominant forms of Se in freshwater. Microalgae act as a major transporter of Se from water to filter-feeders and other organisms. Although most plant species accumulate less than 25 µg Se/g dry weight [72], some microalgae species can accumulate Se at high concentrations (100 µg Se/g dry weight) [73]. Se is essential for many algae and functions to protects them from oxidative damage. Sun et al. [74] indicated that *C. vulgaris* can accumulate Se at high concentrations (857 µg/g dry weight) when grown under Se concentrations of 0–200 mg/L in a growth medium and that relatively low Se concentrations (75 mg selenite/L medium) positively promotes *C. vulgaris* growth and acts as an antioxidant by inhibiting lipid peroxidation and intracellular reactive oxygen species. The maximum accumulation of organic Se was found at 316 µg/g dry weight under relatively low Se (75 mg selenite/L medium) conditions [75], indicating that *C. vulgaris* is an efficient Se accumulator and that Se-enriched *Chlorella* cells might be useful for human supplementation.

### 2.3. Pigments

Carotenoids are secondary metabolites in the most abundant naturally occurring pigments that participate in various biological processes in plants, including photosynthesis, photomorphogenesis, photoprotection, and development [76]. They also serve as colorants and critical components of the human diet, such as antioxidants and provitamin A [76]. More than 400 carotenoids have been identified in living organisms [77], and β-carotene, astaxanthin, lutein, zeaxanthin, and lycopene are widely known as the major carotenoids among them. The green microalgae *Dunaliella salina* produces high concentrations of β-carotene of up to 14% of algal dry weight [78]. *Haematococcus pluvialis* increases astaxanthin concentration up to 4–5% of algal dry weight [79] under stressful conditions. *Chlorella* products contain lower contents of total carotenoids (approximately 1.3%) [80], compared with the above-mentioned green algae. *C. vulgaris* reportedly produces lutein as the primary carotenoid [81,82]. However, *C. zofingiensis* reportedly accumulates significant amounts of astaxanthin, and it might be a suitable organism for the mass production of astaxanthin [83].

## 3. Pharmacological Activities of *Chlorella* Products

Because *Chlorella* cells contain various nutrients and biologically active compounds, the effects of *Chlorella* supplementation on preventing the development of various diseases has been studied in rats and mice, including disease-specific model animals. These animal studies have been useful for elucidating the specific health effects of *Chlorella* supplementation. Moreover, the effects of *Chlorella* supplementation on mitigating a variety of diseases in humans have been investigated. These studies have used either *C. vulgaris* or *C. pyrenoidosa* because these species are commercially available as *Chlorella* products.

### 3.1. Antihypertensive Effects

Hypertension increases the risk of cardiovascular disease [84]. Antihypertensive compounds in foods have been identified using a stroke-prone spontaneously hypertensive (SHRSP) rat model, which is genetically predisposed to hypertension and cerebral stroke [85]. Sansawa et al. [86] investigated the effects of dried *Chlorella* powder (*C. regularis*) on blood pressure, cerebral stroke lesions, and the life span of SHRSP rats. In 12-week-old SHRSP rats fed *Chlorella* (5%, 10%, and 20%) for 13 weeks, elevated blood pressure significantly decreased in the 10% and 20% *Chlorella* groups compared with the untreated controls. Serum total cholesterol levels were significantly lower in all *Chlorella* groups, and their average life span was more than that of the controls. To characterize the antihypertensive compounds in *Chlorella*, *Chlorella* powder was fractionated into hot-water-soluble, lipid-soluble, and residual fractions. Blood pressure was significantly lower in rats fed the lipid or residual fraction but not in those fed the hot-water-soluble fraction. The lipid fraction contained substantial amounts of carotenoids, which are potent antioxidants, and phospholipids, which mediate aorta collagen and elastin metabolism. The residual fraction contained a high level of arginine, which increases the production of endothelium-derived relaxing factor. These beneficial effects of *Chlorella* powder on SHRSP rats might result from synergism between its numerous bioactive compounds.

To evaluate whether daily *Chlorella* supplementation can reduce blood pressure in subjects with mild to moderate hypertension, a pilot study was conducted in 24 participants administered *C. pyrenoidosa* (10 g of *Chlorella* tablets and 100 mL *Chlorella* extract) [87]. After two months of *Chlorella* supplementation, the average heart rate and sitting systolic and diastolic blood pressures only slightly changed. On the other hand, for some subjects with mild to moderate hypertension, *Chlorella* supplementation reduced or maintained their sitting diastolic blood pressure.

Arterial stiffness is a well-established risk factor of cardiovascular disease [88]. Previous studies have reported that antioxidants [89], potassium [90], and *n*-3 unsaturated fatty acids [91] decrease arterial stiffness. Nitric oxide (NO), derived from arginine in the vascular endothelium, is an important modulator of arterial stiffness [92]. *Chlorella* products contain antioxidants, vitamins, potassium, arginine, and *n*-3 unsaturated fatty acids. To evaluate the effects of *Chlorella* supplementation on arterial stiffness, a single-blinded, placebo-controlled crossover study was conducted in 14 young participants who were administered *C. pyrenoidosa* (6 g/day) or placebo for four weeks, with a 12-week washout period between trials, in a randomized order [93]. No differences were observed in blood pressure or heart rate before and after supplementation in both the placebo and *Chlorella* groups. Brachial-ankle pulse wave velocity, a measure of arterial stiffness, decreased in the *Chlorella* group but not in the placebo group [93]. A similar trial in 32 middle-aged and older subjects reports that the brachial-ankle pulse wave velocity decreased after *Chlorella* supplementation but not after placebo supplementation [94]. These changes in brachial-ankle pulse wave velocity with *Chlorella* supplementation correlated with the plasma NOx level. These results suggest that *Chlorella* supplementation decreases arterial stiffness in both younger and older subjects.

The efficacy of *Chlorella* supplementation in reducing cardiovascular risk factors was assessed in a meta-analysis of 19 randomized controlled trials including 797 subjects [95]. This study concluded that *Chlorella* supplementation improves total cholesterol levels, low-density lipoprotein cholesterol levels, systolic blood pressure, diastolic blood pressure, and fasting blood glucose levels but not triglyceride levels, high-density lipoprotein cholesterol levels, and body mass index.

### 3.2. Antihypercholesterolemic and Antihyperlipemic Effects

Elevated total cholesterol and triglycerides and abnormal metabolism of lipoproteins and apolipoproteins are responsible for an increased risk of cardiovascular disease [96,97,98]. The indigestible components of foods, such as dietary fiber, decrease serum cholesterol by inhibiting the intestinal absorption of neutral steroids [99]. *Chlorella* supplementation reportedly decreases serum cholesterol levels in model animals [100]. To identify the bioactive compounds responsible for this effect, the indigestible fraction of *C. regularis* powder was isolated and characterized, revealing a content of 43% crude protein, 37.3% dietary fiber, 6.9% carbohydrate, 5.4% moisture, 4.3% crude fat, and 2.7% ash [101]. Rats fed a diet with 5.3% of this indigestible fraction demonstrated lower serum and liver cholesterol levels and higher fecal neutral steroid levels as compared with those fed a diet of 12.7% *Chlorella* powder. Both *Chlorella* powder and the indigestible fraction exhibited a high bile-acid binding capacity in vitro. Furthermore, the indigestible fraction increased the hepatic mRNA levels of cholesterol 7α-hydroxylase, which is the rate limiting enzyme for cholesterol catabolism and bile-acid synthesis [102]. These results indicate that the indigestible fraction of *Chlorella* possesses hypercholesteromic activity, which improves cholesterol catabolism via the upregulation of hepatic cholesterol 7α-hydroxylase expression.

*Chlorella* supplementation is also reported to decrease serum cholesterol levels in hyperlipemia and mild hypercholesterolemic patients in a small, open-label trial [103]. To evaluate the preventive role of *Chlorella* in maintaining serum cholesterol levels against excess dietary cholesterol intake, a double-blind, randomized, placebo-control study was conducted in 63 mildly hypercholesterolemic subjects treated with either *C. vulgaris* (5 g/day) or placebo for four weeks [104]. A similar trial investigated cholesterol levels in 34 participants administered 510 mg of dietary cholesterol from three eggs concomitantly with either *Chlorella* (*C. vulgaris*) (5 g/day) or a matched placebo for 4 weeks [105]. Participants on the three-egg diet alone exhibited significant elevation in serum total cholesterol, low-density lipoprotein cholesterol, and high-density lipoprotein cholesterol levels. The administration of *Chlorella* in addition to the three-egg diet significantly suppressed these increases in total cholesterol and low-density lipoprotein cholesterol levels and significantly increased serum lutein and α-carotene levels [105]. In mildly hypercholesterolemic subjects, *Chlorella* administration resulted in marked changes in total cholesterol, triglycerides, lutein/zeaxanthin, and α-carotene levels as well as a significant decrease in very low-density lipoprotein cholesterol, apolipoprotein B, non-high-density lipoprotein, and high-density lipoprotein/triglyceride levels [104]. These results suggest that *Chlorella* might inhibit the intestinal absorption of dietary and endogenous lipids. In addition, the observed changes in serum lipids may be associated with changes in serum carotenoids. These results suggest that daily consumption of *Chlorella* provides potential health benefits by reducing the levels of serum lipid risk factors, such as triglycerides and total cholesterol, in mild hypercholesterolemic subjects.

### 3.3. Antidiabetic Effect

Type 2 diabetes, accounting for 90–95% of all diabetes cases, is a severe health problem affecting over 380 million people worldwide [106]. Elevated blood glucose levels, insulin resistance, and low insulin sensitivity are the main characteristics of patients with type 2 diabetes [107], resulting in serious conditions, including arteriosclerosis, renal damage, and retinopathy [108]. In a streptozotocin-induced animal model of diabetes, several studies have been conducted to elucidate the mechanisms underlying the antidiabetic activity of *Chlorella* supplementation [109,110,111]. Shibata et al. [109] evaluated the effects of *Chlorella* supplementation on antioxidant status and cataracts by feeding a diet containing 7.3% (*w*/*w*) *Chlorella* powder (*C*. *regularis*) to 11-week old rats with streptozotocin-induced diabetes. After 11 weeks of supplementation, serum lipid peroxide levels (an index of oxidative status) and blood glycated hemoglobin were lower in *Chlorella*-supplemented rats than in control rats; however, the serum glucose level did not differ between groups. *Chlorella* supplementation delayed the development of lens opacities. These results indicate that *Chlorella* supplementation might be beneficial for preventing diabetes complications such as cataracts, possibly due to the activity of its antioxidant compounds.

Cherng and Shih reported potential hypoglycemic effects of *Chlorella* supplementation in streptozotocin-induced diabetic mice [110]. Oral administration of *Chlorella* 60 min before glucose administration (0.5 g/kg body weight) resulted in a transient hypoglycemic effect at 90 min after glucose administration without an increase in insulin secretion. *Chlorella* supplementation increased 2-deoxyglucose uptake in the liver and soleus muscles of streptozotocin-treated mice and was likely the cause of the observed hypoglycemic effects [111].

The prophylactic effect of *Chlorella* (*C. vulgaris*) supplementation on diabetes was studied by Vecina et al. [112], who investigated body weight, lipid profile, blood glucose, and insulin signaling in liver, skeletal muscle, and adipose tissue in high-fat diet-induced obese mice. *Chlorella* supplementation improves glycemic control in obesity and diabetes because it decreases insulin resistance caused by increased expression of glucose transporter 4 via the activation of protein kinase B phosphorylation in skeletal muscle. *Chlorella* supplementation combined with aerobic exercise training showed more pronounced effects on the improvement of glycemic control via increased activation of muscle phosphorylation signaling in type 2-diabetic rats [113].

A randomized, double-blind, placebo-controlled human study was conducted in 28 borderline-diabetic participants treated with either *Chlorella* (8 g/day) or placebo for 12 weeks [114]. The expression levels of 252 genes, including six associated with type 2 diabetes, differed between the two groups. Notably, the mRNA expression level of resistin, an insulin resistance inducer, was significantly lower in the *Chlorella* group than in the placebo group and correlated with the expression levels of hemoglobin A_1c_, tumor necrosis factor-a, and interleukin-6 [114], all of which are involved in glucose metabolism and/or inflammation.

### 3.4. Hepatoprotective Effect

Li et al. [115] demonstrated that *C. vulgaris* extract has a potent hepatoprotective effect on carbon tetrachloride-induced acute hepatic injury in mice. *Chlorella* extract of 50, 100, or 200 mg/kg of diet, was administered to mice every other day for four weeks, and carbon tetrachloride was administrated intraperitoneally 3 h after the final *Chlorella* supplement. Carbon tetrachloride treatment increased serum alanine and aspartate aminotransferases levels, lipid peroxidation, and cytochrome P450 expression and decrease in reduced glutathione and cellular antioxidant defense enzyme levels; all of these changes were significantly lower in the *Chlorella* (100 and 200 mg/kg diet) groups. Although hepatocyte necrosis was mildly diminished in the 50 mg/kg *Chlorella*-treated group, it was absent in the 100 and 200 mg/kg *Chlorella*-treated groups. These results indicate that *Chlorella* extract has a protective effect on carbon tetrachloride-induced acute hepatic injury in mice, presumably due to the inhibition of carbon tetrachloride-induced cytochrome P450 activation and the activation of antioxidant enzymes and free radical scavengers.

Non-alcoholic fatty liver disease (NAFLD) is a group of metabolic disorders that involving abnormal fat accumulation of more than 5–10% in hepatocytes [116]. It affects 10–35% of the world population [117]. NAFLD includes steatosis, non-alcoholic steatohepatitis, fibrosis, cirrhosis, and hepatocellular carcinoma [118]. Most NAFLD patients have at least one characteristic metabolic syndrome, including insulin resistance, hypertension, dyslipidemia, diabetes, and obesity [119]. Seventy NAFLD patients were randomly administered *C. vulgaris* (1.2 g/day) or placebo for eight weeks [120]. The mean body weight and serum concentrations of liver enzymes were significantly lower in the *Chlorella* group than in the placebo group, and the serum insulin concentration was significantly higher in the *Chlorella* group than in the placebo group. Therefore, *Chlorella* supplementation may have beneficial effects on reducing weight and serum glucose levels and improving inflammatory biomarkers as well as liver function in NAFLD patients [120,121].

To evaluate the safety and efficacy of *Chlorella* (*C. pyrenoidosa*) in patients chronically infected with hepatitis C virus genotype 1, patients received daily oral supplement of *Chlorella* (both of *Chlorella* extract and tablets) for 12 weeks [122]. The majority (approximately 85%) of the patients exhibited a significant decrease in alanine aminotransferase levels from Week 0 to Week 12. Patients with decreased alanine aminotransferase level showed a tendency toward decreased hepatitis C virus load.

### 3.5. Detoxification Effect

Dioxins are a group of polychlorinated dibenzo-*p*-dioxin and dibenzofuran-related compounds that are industrial contaminants and ubiquitous environmental pollutants [123]. These compounds are easily absorbed in the mammalian gastrointestinal tract [124] and then stored in the liver, adipose tissue, and breast milk due to their lipophilic properties [125]. An incident involving the consumption of cooking oil contaminated with dioxins had tragic effects [126]. To investigate the effects of *Chlorella* supplementation on fecal excretion of dioxins, rats were administered dioxin-contaminated rice oil [127]. The rats were fed 4 g of a 10% (*w*/*w*) *Chlorella* (*C. vulgaris*) diet or a control diet (without *Chlorella*) once during the five-day experimental period, and the amounts of fecal dioxins were measured. The fecal dioxin levels were significantly greater in the *Chlorella* group than in the control group. In addition, *Chlorella* supplementation significantly inhibited the gastrointestinal absorption of dioxins (approximately 2–53% decrease). These results indicate that *Chlorella* supplementation might be useful in promoting dioxin excretion.

Heterocyclic amines have been established as carcinogenic chemicals that form when amino acids, sugars, and creatine in muscle meats (beef, pork, fish, and poultry) react with one another during cooking at high temperatures [128]. To evaluate the effect of *Chlorella* supplementation on the detoxification of carcinogenic heterocyclic amines, a randomized, double-blind, placebo-controlled crossover study with *Chlorella* supplementation (100 mg/day) for two weeks was conducted [129]. *Chlorella* supplementation decreased urinary excretion of the predominant metabolite of carcinogenic heterocyclic amines [129], suggesting that *Chlorella* either inhibits the intestinal absorption of heterocyclic amines or inactivates carcinogenic compounds.

Methylmercury is a neurotoxic metal compound that is converted from inorganic mercury by microorganisms in aquatic environments and is then accumulated in fish and shellfish through marine food chains [130]. Therefore, the major route of human exposure to methylmercury is the consumption of seafood [130]. In many countries, pregnant women are cautioned against consuming large fish, such as tuna, to prevent fetal exposure [131]. As *Chlorella* consumption is reported to increase the excretion of methylmercury and lower tissue mercury levels in methylmercury-treated mice [132], an open-label clinical trial was performed to estimate the effects of *Parachlorella beijerinckii* supplementation (9 g/day) for three months on mercury concentrations in the hair and blood of healthy subjects [133]. *Chlorella* supplementation reduced mercury levels in both the hair and blood [133]. Fecal excretion is the major route of methylmercury elimination (90%) in humans [134]. Most of the methylmercury in the liver is secreted as a glutathione complex via the bile duct, with a small portion excreted in the feces [135]. The dietary fiber in *Chlorella* cells increases the amount of feces excreted by humans [136]. Dietary fiber has been shown to absorb some methylmercury in vitro [132]. These observations suggest that the observed lowering of hair and blood mercury levels in *Chlorella*-treated participants may result from the promotion of fecal methylmercury excretion via accelerated bile secretion, the binding of methylmercury to dietary fiber in the intestinal tract, and increased feces production.

### 3.6. Immunomodulatory Effects

Allergic disease is a prevalent aberrant immune responsive against innocuous environmental proteins (antigens) [137]. Allergen-specific CD4^+^ T cells involved in the initiation of allergic reactivity can develop into either type 1 or type 2 helper T cells [138]. CD4^+^ T cells stimulated in the presence of interleukin-12 and γ-interferon can develop into type 1 helper T cells [138], while interleukin-4 promotes the development of type 2 helper T cells and inhibits the generation of type 1 helper T cells [139]. Since type 1 and 2 helper T cells regulate each other, interleukin-12 functions not only to induce the type 1 helper T-cell response but also to regulate the type 2 helper T-cell response [140]. Interleukin-12 strongly suppresses the production of IgE by preventing type 2 helper T-cell development [141]. Allergen-specific IgE induces the pathogenesis of allergic disorder [142].

Hasegawa et al. [143] described the effects of a *Chlorella* (*C. vulgaris*) hot-water extract on antigen specific response in mice. A 2% (*w*/*w*) *Chlorella* hot-water extract diet or control diet (without *Chlorella* extract) was given to mice for two weeks before intraperitoneal administration of casein/complete Freund’s adjuvant (an immunostimulant). Mice that received the hot-water extract exhibited suppressed IgE production and mRNA expression of interleukin-6 involved in the type 2 helper T-cell response. They also exhibited increased levels of interleukin-12 and g-interferon mRNA, increasing the type 1 helper T-cell response and suppressing the type 2 helper T-cell response. These results suggest that *Chlorella* hot-water extract supplementation might be useful for suppressing allergic responses with a predominant type 2 helper T-cell response. To clarify the mechanisms underlying the immunomodulatory activity of *Chlorella* hot-water extract, soluble polysaccharides were isolated from *C. pyrenoidosa* hot-water extract and characterized [144]. GC-MS analysis indicated that the major monosaccharide components of the soluble polysaccharides are rhamnose (31.8%), glucose (20.4%), galactose (10.3%), mannose (5.2%), and xylose (1.3%). These soluble polysaccharides were intraperitoneally administrated (100 mg/kg of body weight) to 6–8-week-old mice. After 24 h, lipopolysaccharide as an antigen was administrated to mice, and their serum was collected 1.5 h later [144]. The soluble polysaccharides induced interleukin-1b secretion in macrophages via the toll-like receptor protein kinase signaling pathway. Interleukin-1β is one of the most important mediators of inflammation and host responses to infection [145]. These results suggest that *Chlorella* hot-water-soluble polysaccharides could be used as an agent source to stimulate anti-microorganism activity.

Halperin et al. [146] evaluated the effect of *C. pyrenoidosa* supplementation (200 or 400 mg) on the immune response to influenza vaccination. After 28 days of *Chlorella* supplementation, the antibody response to the influenza vaccine was not elevated in the overall study population but was increased in participants aged 50–55 years.

Salivary secretory immunoglobulin A (SIgA) plays a crucial role in mucosal immune function and is the first line of defense against pathogenic microbial invasion in human [147]. To evaluate whether *Chlorella* supplementation increases salivary SIgA secretion in humans, a blind, randomized, crossover study was conducted in participants administered *Chlorella* (*C. pyrenoidosa*) (6 g/day) or placebo for four weeks [148]. Although no difference was observed in salivary SIgA levels before and after placebo ingestion, salivary SIgA levels were significantly elevated after *Chlorella* ingestion than at baseline. The SIgA secretion rate increased significantly after *Chlorella* supplementation. These results suggest that four-week *Chlorella* supplementation increases salivary SIgA secretion and improves mucosal immune function in humans.

Natural killer cells are the predominant innate lymphocyte subsets that mediate antitumor and antiviral responses [149]. To evaluate the effect of *Chlorella* supplementation on natural killer cell activity and early inflammatory response in humans, a randomized, double-blinded, placebo-controlled trial was conducted in healthy adults ingested with *Chlorella* (*C. vulgaris*) (5 g/day) or placebo [150]. After eight weeks of supplementation, serum interferon-γ and interleukin-1β levels were significantly elevated and that of interleukin-12 tended to increase in the *Chlorella* group. Natural killer cell activities were significantly elevated in the *Chlorella* group. These results suggest a beneficial immunostimulatory effect of short-term *Chlorella* supplementation that increases natural killer cell activity and produces interferon-γ, interleukin-12, and interleukin-1β.

### 3.7. Antioxidant Effects

*C. vulgaris* hot-water extract [151] and acetone extract [152] are reported to have antitumor activity. A *Chlorella* aqueous extract containing substantial amounts of antioxidants also exhibit antiproliferative activity in human hepatoma cells [153]. Lipophilic pigments, including carotenoids antheraxanthin, zeaxanthin, and lutein, extracted from *Chlorella* cells were observed to significantly inhibit the growth of human colon cancer cells [154]. These results suggest that the antitumor activity of *Chlorella* might be the synergistic effect of multiple bioactive compounds. Romos et al. [155] reported that *Chlorella* supplementation can modulate immunomyelopoietic activity and disengage tumor-induced suppression of various cytokines and related cell activities in tumor-bearing mice. Interestingly, a 63.1-kD antitumor glycoprotein was isolated from the culture supernatant of *C. vulgaris* strain CK22 [156,157], and its chemical and antitumor properties were characterized [158], suggesting possible contribution of this glycoprotein toward the observed antitumor activity.

Alzheimer’s disease is a severe neurodegenerative condition affecting humans [159]. The erythrocytes of Alzheimer’s disease patients are known to be in an excessively oxidized state [160]. α-Tocopherol and carotenoids such as lutein are important lipophilic antioxidants in human erythrocytes [161]. Erythrocyte lutein levels were found to be significantly lower in Alzheimer’s disease patients than in normal subjects [162]. Oral intake of lutein capsules increases lutein levels and prevents phospholipid hydroperoxide accumulation in human erythrocytes [163], suggesting that dietary lutein has the potential to act as an important antioxidant in erythrocytes and thus may have beneficial effects in Alzheimer’s disease patients. According to the labels on *Chlorella* products D and M, the products contain substantial amounts of lutein (approximately 200 mg/100 g dry weigh). A randomized, double-blind, placebo-controlled human study was conducted to evaluate the effects of *Chlorella* supplementation (8 g *Chlorella*/day/person; equivalent to 22.9 mg lutein/day/person) on phospholipid hydroperoxide and lutein levels in erythrocytes [164]. After two months of *Chlorella* supplementation, erythrocyte lutein levels increased 4.6-fold, but tocopherol levels did not change [164], suggesting that daily *Chlorella* intake may be effective for improving and maintaining erythrocyte antioxidant status and lutein levels in humans. These results suggest that *Chlorella* supplementation contributes to maintaining the normal function of erythrocytes and has beneficial effects on Alzheimer’s disease-related dementia in humans.

Major depressive disorder is a widespread mental disorder that greatly impairs the quality of life of humans. Approximately 12% of people experience at least one episode of depression during their lifetime [165]. Although various antidepressant drugs are available for treating depression, a considerable proportion of patients are not responsive to these drugs and some experience side effects [166,167]. Therefore, alternative antidepressant drugs with adequate efficacy and safety are needed. The therapeutic effect of dried *C. vulgaris* extract administration (1.8 g/day) for six weeks was evaluated in patients with major depressive disorder [168]. After treatment, the participants exhibited improvements in physical and cognitive symptoms of depression [168]. As oxidative stress is an important pathophysiological mechanism underlying major depressive disorder, major depressive disorder has been effectively reversed via antioxidant therapy [169,170]. These observations suggest that the therapeutic effectiveness of *Chlorella* supplementation may result from the action of its antioxidant nutrients and compounds [171].

### 3.8. Other Effects

Stress is well known to disturb homeostasis, impairing immunological functions. *Chlorella* supplementation reportedly stimulates the pool of hematopoietic stem cells and activates leukocytes [172]. To further understand the influence of *Chlorella* (*C. vulagaris*) supplementation on hematopoiesis, hematopoietic cell populations in the bone marrow of mice subjected to a single or repeated stressor were measured [173]. Reduced numbers of hematopoietic progenitors in the bone marrow were observed after treatment with either stressor. Both stressors induced a decrease in mature myeloid and lymphoid populations but did not affect pluripotent hematopoietic progenitors. Both stressors reduced the levels of interleukin-1α and interleukin-6. *Chlorella* supplementation prevented the changes produced by both stressors in all of the parameters tested, suggesting that *Chlorella* supplementation is an effective tool for the prophylaxis of myelosuppression caused by single or repeated stressors.

Stressors are processed in the brain through the activation of several types of neurons. Immediate early genes such as *c-fos* are extensively used to map brain areas involved in stress responses [174]. Using *c-fos* expression, Oueiroz et al. [175] evaluated the effect of acute pretreatment with *Chlorella* (*C. vulgaris*) on the peripheral and central responses to forced swimming stress in rats. *Chlorella* supplementation produced a significant reduction in stress-related hypothalamic–pituitary–adrenal axis activation due to decreased corticotrophin releasing factor gene expression in the hypothalamic paraventricular nucleus and a lower adrenocorticotropic hormone response. Hyperglycemia induced by the stressor was similarly reduced. These results suggest that *Chlorella* supplementation might reduce the impact of stressors.

A hot-water extract of *Chlorella* (*C. pyrenoidosa*) increased the lifespan of superoxide dismutase-1 mutant adults of *Drosophila melanogaster* in a dose-dependent manner (200–800 µg/mL) [176]. An active compound was purified and identified as phenethylamine, an aromatic amine, which exhibited no superoxide dismutase-like activity. Treatment with this compound extended the lifespan of the mutant flies at very low concentration (60 ng/g diet) [176]. Similarly, supplementation of *C. sorokiniana* (4 mg/mL) reportedly increased the lifespan of *D. melanogaster* by 10% increase as compared to a control diet, likely due to the increased mRNA expression of antioxidative enzymes (Cu/Zn-superoxide dismutase and catalase) [177].

However, for the beneficial effects described above, no human study has been conducted.

## 4. Conclusions

Commercially available *Chlorella* products contain a variety of nutrients essential for humans, as well as a large amount of good quality protein, dietary fibers, and polyunsaturated fatty acids, including α-linolenic and linoleic acids. In particular, *Chlorella* products contain vitamins D_2_ and B_12_, which are absent from plant-derived food sources, and larger amounts of folate and iron than other plant-derived foods. Mounting scientific evidence of the health benefits of daily *Chlorella* consumption has been presented in animal and human studies. The pharmacological activities reported in *Chlorella* studies include immunomodulation, antioxidative activity, and effects against diabetes, hypertension, and hyperlipidemia. The beneficial effects of *Chlorella* might involve synergism between multiple nutrient and antioxidant compounds. Overall, the information regarding bioactive compounds in *Chlorella* is limited. Thus, new bioactive compounds responsible for its pharmacological activities may be identified in future studies.

## Figures and Tables

**Figure 1 nutrients-12-02524-f001:**
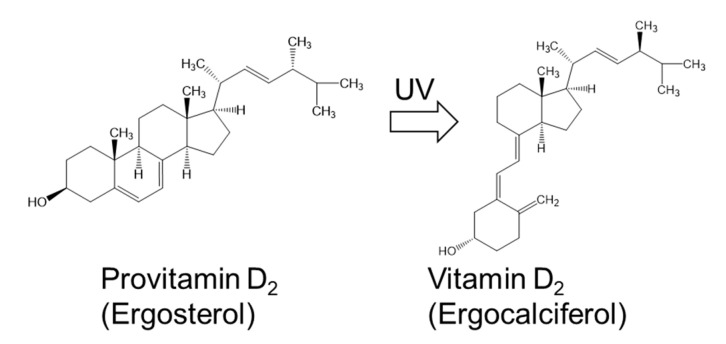
Structures of provitamin D_2_ and vitamin D_2_ found in commercially available *Chlorella* products.

**Figure 2 nutrients-12-02524-f002:**
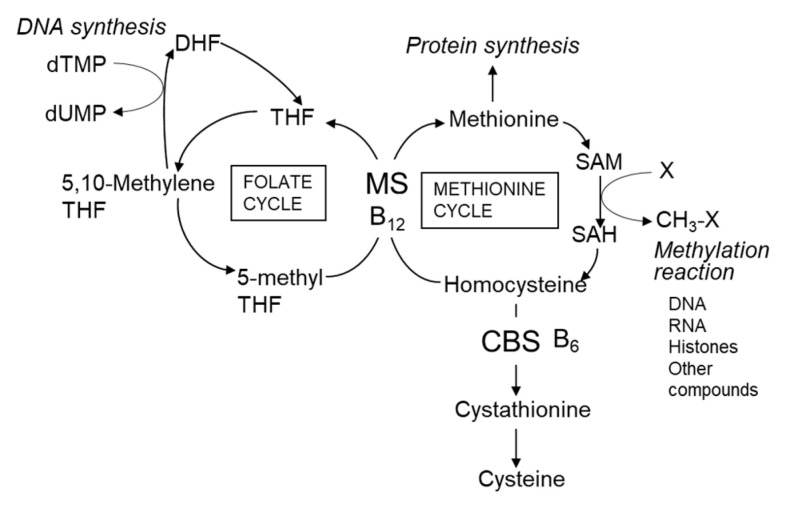
Homocysteine metabolic pathway in mammals. Abbreviations: B_6_, vitamin B_6_; B_12_, vitamin B_12_; CBS, cystathionine β-synthetase; DHF, dihydrofolate; MS, cobalamin-dependent methionine synthase; SAM, S-adenosyl methionine; SAH, S-adenosyl homocysteine; THF, tetrahydrofolate.

**Figure 3 nutrients-12-02524-f003:**
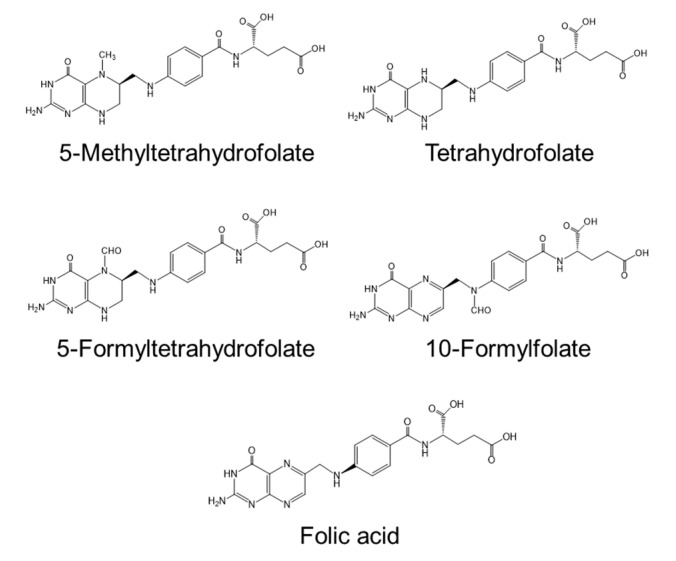
Chemical structures of folate compounds found in commercially available *Chlorella* products.

**Figure 4 nutrients-12-02524-f004:**
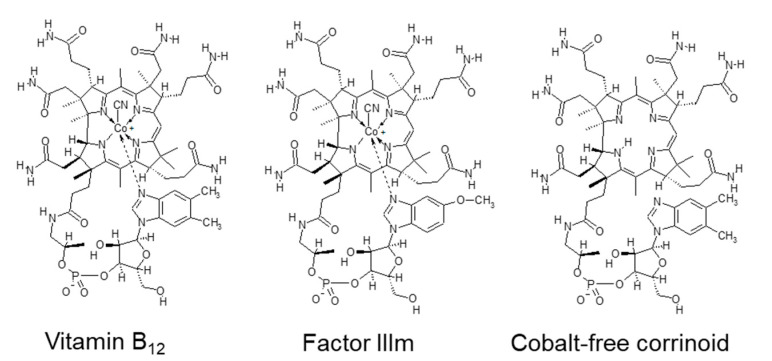
Chemical structures of vitamin B_12_ and related compounds found in commercially available *Chlorella* products. Abbreviations: Factor IIIm, 5-methoxybenzimidazolylcobamide.

**Table 1 nutrients-12-02524-t001:** Nutrient content of 13 commercially available *Chlorella* products.

Macronutrients (Per 100 g Dry Weight)	A	B	C	D	E	F	G	H	I	J	K	L	M
Proteins (g)	50–65	61	63	65	57	50	56–72	50–67	50–70	62	60	58	57
Fats (g)	7–14	10	13	12	11		7–20	5–15	8–15	11	10	10	12
Carbohydrates (g)			15	11			5–23	8–42	8–20		20	18	
Sugars (g)	5–21	7	5	0–1			0–5	2–23		1–10			11
Dietary fibers (g)	7–14	11	10	11	11		5–18	7–18		8–16			10
Remarks	*^1^ *^2^ 78%	*^1^ *^4^	*^2^	*^2^			*^1^ *^2^ 77–82% *^3^	*^2^ 75–85% *^3^	*^1^ *^2^ 83% *^3^	*^2^ 82% *^3^		*^4^	*^1^ *^3^ *^4^

*^1^ Cell walls disrupted; *^2^ digestibility of proteins; *^3^ contains *Chlorella* extract; *^4^
*C. pyrenoidosa.*

**Table 2 nutrients-12-02524-t002:** Amino acid content of commercially available *Chlorella* products C and M.

Amino Acids (mg/100 g Dry Weight)	C	M
Essential		
Isoleucine	1820	2030
Leucine	4180	4480
Lysine	4659	3140
Methionine	1009	1240
Phenylalanine	2230	2580
Threonine	2209	2490
Tryptophan	1030	1090
Valine	2780	3090
Histidine	1141	1040
Non-essential		
Tyrosine	1720	1940
Cystine	659	650
Aspartic acid	4469	4710
Serine	1930	2120
Glutamic acid	6209	6030
Proline	2320	2560
Glycine	2859	2990
Alanine	4009	4170
Arginine	3109	3260

**Table 3 nutrients-12-02524-t003:** Content of vitamins and related compounds in 13 commercially available *Chlorella* products.

Vitamins (Per 100 g Dry Weight)	A	B	C	D	E	F	G	H	I	J	K	L	M
Vitamin B_1_ (mg)		1.9	2.5		6.5		1.0–3.0		1.0–3.0	1.8			1.6
Vitamin B_2_ (mg)	3–8	5.6	5.0	5.7	5.5		2.0–9.0	4.0–9.0	4.0–8.0	5.0		5.0	4.8
Vitamin B_6_ (mg)		0.9	2.5		1.7		1.0–3.0		1.0–3.0	1.0–3.0			1.8
Vitamin B_12_ (μg)		20.0							6.0–30.0	200.0–500.0			230.0
Niacin (mg)		20.4	50.0				40.0–80.0		20.0–50.0	10.0–40.0			45.9
Folate (mg)		0.3	2.0				1.2–3.6						1.4
Biotin (μg)													227.0
Pantothenic acid (mg)										1.0–6.0			1.8
Vitamin C (mg)		7.0	50.0		30.0		10.0–200.0						14.0
Vitamin D_2_ (mg)													1.4
Vitamin E (mg)		3.0	25.0				10.0–45.0						6.2
Vitamin K (mg)		1.4	1.1	0.3						0.5–3.5 *^1^			1.2 *^1^
Carotenoids (mg)				25.0 *^2^			36.0–150.0 *^2^			100.0–500.0			31.5 *^3^

*^1^ Vitamin K_1_ (mg), *^2^ β-Carotene (mg), *^3^ α-Carotene + β-Carotene (mg).

**Table 4 nutrients-12-02524-t004:** Mineral content of 13 commercially available *Chlorella* products.

Minerals (Per 100 g Dry Weight)	A	B	C	D	E	F	G	H	I	J	K	L	M
Sodium (mg)	5–75	65			40		80–220	10–45		5–30	80	65	47
Iron (mg)	10–130	160	121	62			350–1600		100–200	50–100		110	113
Calcium (mg)		650	513		850		500–1500		100–300				433
Potassium (mg)		970	1075		350		200–500			500–1500			1020
Magnesium (mg)		350	250							23–420			298
Zinc (mg)		2											1
Copper (mg)		1											1
Phosphorus (mg)		1600											1320
Manganese (mg)													5

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
