# Peer review of "Potential of Chlorella as a Dietary Supplement to Promote Human Health"

_nutrients, 2020, doi:10.3390/nu12092524_

Round 1

Reviewer 1 Report

In my opinion your work is not reliable impartial scientific analysis. You cited only sources (in addition - little) that fit into the general theory that chlorella is a good supplement. 

Abstract:
It is rather like a post on a social networking site, not an abstract of a scientific review article. 

Introduction:
It is scientifically unimportant popular science information. Here you should indicate what so far the world of science has verified in the context of Chlorella. What your article will contribute to this, how to supplement this knowledge. The aim of the work is laconic and says nothing about what was actually presented. It should be completed and extended.

Part 2.1.

L.59 Relying on the manufacturer's leaflet is always not good.

L.82 "extremely high concentrations" - What does it mean? Compared to what?

L.88-92 are unnecessary

L.105-109 Saying about food as a source of vitamin D is a complete misunderstanding, especially in the context of plant sources. How much sundried mushrooms or Chlorella should be eaten to meet the RDA?

Figure 3 - Different font size

Part 3. - You often cite only one source in support of each thesis (in addition, often on such a small sample that it is difficult to consider it the only correct truth). In one case only you cite the opposing article. This is absolutely unacceptable. 

3.1.3 - Antihypercholesterolemia ?

Figure 5 - this figure adds nothing and in my opinion is completely unnecessary

3.2.4. - this part I could consider as fairly well presented.

3.2.7.-3.2.9 - when discussing the results in many places, research information is laconic and does not provide such important information as the sample size

In the part discussing human studies, information on animal studies is also cited (ex. L. 468). It's inconsistent. And in general I am against such a division into separate discussion of animal and human research. The combination would give you the chance to track all effects and would not lead to a total skipping reading of the part about animal research.

Author Response

Dear Reviewer 1,

Thank you very much for your decision letter of 15th, July 2020, with regard to our manuscript (Nutrients-830452) with the comments from yourself. We appreciate the comments, which are very helpful. We have tried to revise the manuscript in line with suggestions.

In response to comments from reviewer 1, the following changes were made.

  1. In my opinion your work is not reliable impartial scientific analysis. You cited only sources (in addition - little) that fit into the general theory that chlorella is a good supplement.

Ans: We did not select only good results in Chlorella supplementation. We reviewed all papers found out in PubMed database.

  1. Abstract: It is rather like a post on a social networking site, not an abstract of a scientific review article.

Ans: Abstract has been rewritten in the line of your suggestion.

  1. Introduction: It is scientifically unimportant popular science information. Here you should indicate what so far the world of science has verified in the context of Chlorella. What your article will contribute to this, how to supplement this knowledge. The aim of the work is laconic and says nothing about what was actually presented. It should be completed and extended.

Ans: Introduction section has been rewritten in the line of your suggestion.

  1. Part 2.1. L.59 Relying on the manufacturer's leaflet is always not good.

Ans: In general, the data of the manufacture’s leaflets have been determined by certain food analytical companies. Additional data and references have been added in Lines70-71,86,92-98.

L.82 "extremely high concentrations" - What does it mean? Compared to what?

Ans: Additional data and sentence have been added in Lines107-108.

L.88-92 are unnecessary

Ans: These sentences have been delated according to the reviewer’s suggestion.

L.105-109 Saying about food as a source of vitamin D is a complete misunderstanding, especially in the context of plant sources. How much sundried mushrooms or Chlorella should be eaten to meet the RDA?

Ans: Sentences have been revised according to the reviewer’s suggestion (Lines 129-131).

Figure 3 - Different font size

Ans: Font size in Figure 3 has been revised according to the reviewer’s suggestion.

Part 3. - You often cite only one source in support of each thesis (in addition, often on such a small sample that it is difficult to consider it the only correct truth). In one case only you cite the opposing article. This is absolutely unacceptable.

Ans: We did not select only good results in Chlorella supplementation. We reviewed all papers found out in PubMed database. However, in some case, we had not found out one paper in the data base.

3.1.3 - Antihypercholesterolemia ?

Ans: We did not understand your question.

Figure 5 - this figure adds nothing and in my opinion is completely unnecessary

Ans: Figure 5 has been deleted according to the reviewer’s suggestion.

3.2.4. - this part I could consider as fairly well presented.

3.2.7.-3.2.9 - when discussing the results in many places, research information is laconic and does not provide such important information as the sample size

In the part discussing human studies, information on animal studies is also cited (ex. L. 468). It's inconsistent. And in general I am against such a division into separate discussion of animal and human research. The combination would give you the chance to track all effects and would not lead to a total skipping reading of the part about animal research.

Ans: Sections of animal and human studies and have been rewritten in one section according to the reviewer’s suggestion.

Reviewer 2 Report

I think that it is worthwhile having a review of Chlorella in human health. I like the layout, it is easy to follow, progresses logically and, in my opinion, importantly separates animal from human studies. I do feel that there is a bit of a missed opportunity to mention (briefly): 1.The flexibility of chlorella as a healthful product (different growth conditions change macro, micro and secondary metabolite make up) 2. Discuss Selenium and carotenoid levels. 3. Mention a 2019 review on general micro algae and heath that was published in marine drugs as a source of information on other micro algae and health (e.g here we focus on Chlorella, a general review on micro algae was recently published here.)

"Microalgae for High-Value Products Towards Human Health and Nutrition"

General Comments

During the nutrients section I found myself asking, which species of Chlorella is this referring and are the differences between the species really meaningful? The species data is sometimes presented however is absent in table 1. Please make sure it is clear what species is being discussed throughout the review. I would also have liked to see mention of selenium content. This is an important micronutrient in many countries, relates to later discussed measure of oxidative stress, and it bioaccumulates in Chlorella. See Sun eta al 2014. Selenium Accumulation in Unicellular Green Alga Chlorella vulgaris and Its Effects on Antioxidant Enzymes and Content of Photosynthetic Pigments. Also within the minerals section, there is listed Carotenoids within table 3, but there is no real mention of them in the text of that section. To this point, I also feel there is a missed opportunity to further explore the potential of Chlorella as a flexible nutraceutical product.   This is very much a status quo review, how about some forward looking directions? Chlorella is a flexible organism, it can have macro and micronutrient content meaningfully altered by changes to growth conditions. Or, mention this flexibility in the Nutrients section, highlighting the possible role of different species and different growth conditions and how it may relate to improved/diverse health application.

e.g Carbohydrate content, (See Growth, nutrient uptake and chemical composition of Chlorella sp. and Nannochloropsis oculata under nitrogen starvation)

Carotenoid content (see review Microalgae as a Feedstock for Biofuel Precursors and Value-Added Products: Green Fuels and Golden Opportunities).

Specific comments

Abstract/Introduction

line 10. I would add that Chlorella is a green single celled (unicelluar) algea….which is commercially produced etc. 

line 24 25 starting 2 sentences with "since" reads strangely, suggest a rework of those 2 sentences. 

line 30-31. I would remove the reference to population growth. There are many causes for famine.

Nutrients

lines 59-61. Information about cell wall indigestibility should be discussed at the start of this section, probably after 'Table 1" on line 44. E.g Chlorella is rich in various proteins, minerals….etc, however humans cannot digest…. etc..

Line 71. I do not believe there is good evidence for Arachidonic acid being heart healthy (mixed at best), suggest you remove. In fact, you could reword this paragraph to be tighter, as you mention EPA, DHA and Arachidonic acid, then say they are not in chlorella, could just mentions the heart healthy ones that are in Chlorella.

Pharmacological

Line 189. Mention of antioxidant effects, these could also be related to selenium consumption, another reason to discuss selenium

Lines 220 to 222. Sentence is worded strangely, makes it read like the rats life span was extended at 21 weeks.

Line 234, Information about species of Chlorella would be nice.

Lines 260-261. This paragraph seems to stick out bit on its own. Suggest you join to the main last paragraph. E.g Interestingly, a 63.1 KD…….. suggesting possible contribution of this glycoprotein to the anti-tumorigenic properties observed.  

Line 274. Faecal dioxin levels were 0.2 to 13 times greater is confusing without context given in the paper. Suggest you present this data as an average of the total dioxins, not as a range for the approx. 20 different dioxins measured in to papers. E.g approx. 4 times greater faecal levels etc. You will need to average the values from the paper yourself.

Line 352 I take it that these 2 species of Chlorella make up the in the chlorella tested in the earlier section?

Line 538. I would take another look at the Author contributions, seems a bit weird for a review. 

Author Response

Dear Reviewer 2,

Thank you very much for your decision letter of 15th, July 2020, with regard to our manuscript (Nutrients-830452) with the comments from yourself. We appreciate the comments, which are very helpful. We have tried to revise the manuscript in line with suggestions.

In response to comments from reviewer 2, the following changes were made.

General Comments

During the nutrients section I found myself asking, which species of Chlorella is this referring and are the differences between the species really meaningful? The species data is sometimes presented however is absent in table 1. Please make sure it is clear what species is being discussed throughout the review.

Ans: We did not get information of Chlorella species from the labels or leaflets of the commercially available products shown in Table 1. However, we have tried to described Chlorella species in the text.

I would also have liked to see mention of selenium content. This is an important micronutrient in many countries, relates to later discussed measure of oxidative stress, and it bioaccumulates in Chlorella. See Sun eta al 2014. Selenium Accumulation in Unicellular Green Alga Chlorella vulgaris and Its Effects on Antioxidant Enzymes and Content of Photosynthetic Pigments. Also within the minerals section, there is listed Carotenoids within table 3, but there is no real mention of them in the text of that section.

Ans: Some sentences regarding selenium and carotenoids have been added in Lines 199-215 and Lines 216-228, respectively, according to the reviewer’s suggestion.

 To this point, I also feel there is a missed opportunity to further explore the potential of Chlorella as a flexible nutraceutical product.   This is very much a status quo review, how about some forward looking directions? Chlorella is a flexible organism, it can have macro and micronutrient content meaningfully altered by changes to growth conditions. Or, mention this flexibility in the Nutrients section, highlighting the possible role of different species and different growth conditions and how it may relate to improved/diverse health application.

e.g Carbohydrate content, (See Growth, nutrient uptake and chemical composition of Chlorella sp. and Nannochloropsis oculata under nitrogen starvation)

Carotenoid content (see review Microalgae as a Feedstock for Biofuel Precursors and Value-Added Products: Green Fuels and Golden Opportunities).

Ans: Some sentences have been added in Lines 99-101 according to the reviewer’s suggestion.

Specific comments

Abstract/Introduction

line 10. I would add that Chlorella is a green single celled (unicelluar) algea….which is commercially produced etc.

Ans: The sentence has been revised according to the reviewer’s suggestion (Line 10).

line 24 25 starting 2 sentences with "since" reads strangely, suggest a rework of those 2 sentences.

Ans: Some sentences have been revised according to the reviewer’s suggestion (Line 40-42).

line 30-31. I would remove the reference to population growth. There are many causes for famine.

Ans: The reference has been deleted according to the reviewer’s suggestion (Line 47).

Nutrients

lines 59-61. Information about cell wall indigestibility should be discussed at the start of this section, probably after 'Table 1" on line 44. E.g Chlorella is rich in various proteins, minerals….etc, however humans cannot digest…. etc..

Ans: Sentences have been revised according to the reviewer’s suggestion (Lines 63-66).

Line 71. I do not believe there is good evidence for Arachidonic acid being heart healthy (mixed at best), suggest you remove. In fact, you could reword this paragraph to be tighter, as you mention EPA, DHA and Arachidonic acid, then say they are not in chlorella, could just mentions the heart healthy ones that are in Chlorella.

Ans: The sentences pointed out by the reviewer have been deleted.

Pharmacological

Line 189. Mention of antioxidant effects, these could also be related to selenium consumption, another reason to discuss selenium

Ans: Some sentences regarding selenium have been added in Lines 199-215.

Lines 220 to 222. Sentence is worded strangely, makes it read like the rats life span was extended at 21 weeks.

Ans: The sentence pointed out by the reviewer has been revised (Lines 244-245).

Line 234, Information about species of Chlorella would be nice.

Ans: Chlorella species have been added in Line 286.

Lines 260-261. This paragraph seems to stick out bit on its own. Suggest you join to the main last paragraph. E.g Interestingly, a 63.1 KD…….. suggesting possible contribution of this glycoprotein to the anti-tumorigenic properties observed.

Ans: Some sentences have been revised according to the reviewer’s suggestion (Lines 480-482).

Line 274. Faecal dioxin levels were 0.2 to 13 times greater is confusing without context given in the paper. Suggest you present this data as an average of the total dioxins, not as a range for the approx. 20 different dioxins measured in to papers. E.g approx. 4 times greater faecal levels etc. You will need to average the values from the paper yourself.

Ans: We have tried to revise the sentences according to the reviewer’s suggestion (Lines 391-394).

Line 352 I take it that these 2 species of Chlorella make up the in the chlorella tested in the earlier section?

Ans: Sections of animal and human studies and have been rewritten in one section according to the reviewer I’s suggestion.

Line 538. I would take another look at the Author contributions, seems a bit weird for a review. 

Ans: Sentences have been revised according to the reviewer’s suggestion (Lines 554-555).

Round 2

Reviewer 1 Report

The authors made corrections however still do not cite all the sources they could. This is not a fair review article suitable for the journal with such a high IF.